

# Spatio-temporal optimization of groundwater monitoring networks using data-driven sparse sensing methods

Marc Ohmer[1], Tanja Liesch[1], and Andreas Wunsch[1]

[1]Institute of Applied Geosciences, Division of Hydrogeology, Karlsruhe Institute of Technology, Karlsruhe, Germany

**Correspondence:** Marc Ohmer (marc.ohmer@kit.edu)

**Abstract.** Groundwater monitoring and specific collection of data on the spatio-temporal dynamics of the aquifer are pre-requisites for effective groundwater management and determine nearly all downstream management decisions. An optimally designed groundwater monitoring network will provide the maximum information content at the minimum cost (Pareto opti-mum). In this study, *PySensors*, a Python package containing scalable, data-driven algorithms for sparse sensor selection and

signal reconstruction with dimensionality reduction is applied to an existing groundwater monitoring network (GMN) in 1D (hydrographs) and 2D (gridded groundwater contour maps). The algorithm first fits a basis object to the training data, then applies a computationally efficient QR algorithm that ranks existing monitoring wells (for 1D) or suitable sites for additional monitoring (for 2D) in order of "importance" based on the state reconstruction to this tailored basis. This procedure enables a network to be reduced or extended along the Pareto front. Moreover, we investigate the effect of basis choice on reconstruction

performance by comparing three types typically used for sparse sensor selection (identity, random projection, and singular value decomposition resp. principal component analysis). We define a gridded cost function for the extension case penalizes unsuitable locations. Our results show that this approach is generally better than the best randomly selected wells.The opti-mized reduction makes it possible to adequately reconstruct the removed hydrographs with a highly reduced subset with low loss. An average absolute reconstruction accuracy of 0.1 m is achieved with a subset of 6% wells, 0.05 m with 31%, and 0.01

m with 82% wells.

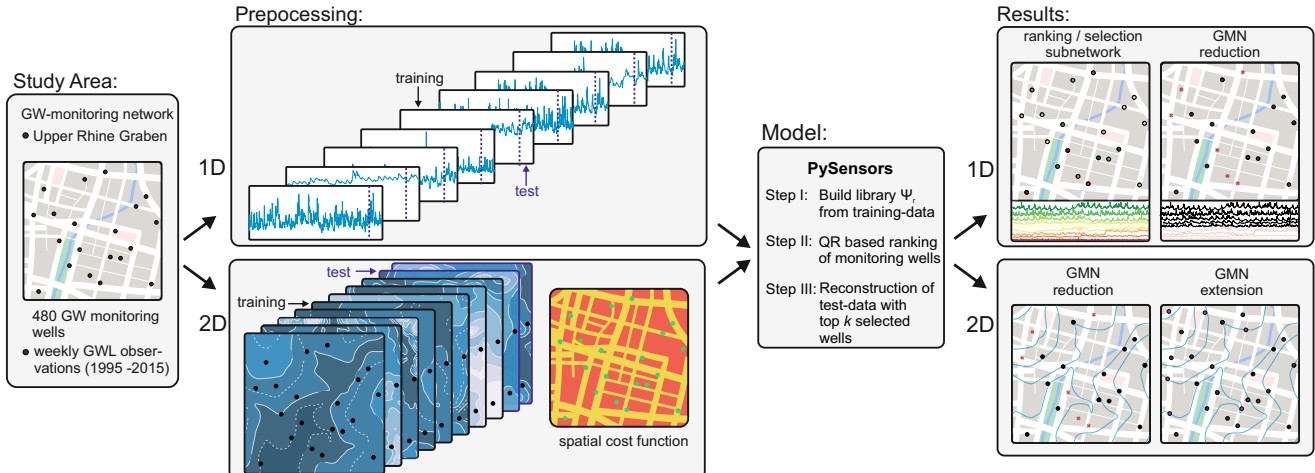



# 1   INTRODUCTION

Groundwater is a vital resource for drinking water supply and industrial, commercial, as well as agricultural uses. Therefore, effective groundwater management and monitoring practices are critical to ensure the availability and quality of water supplies for future generations. A groundwater monitoring network (GMN) is defined by a spatial arrangement of groundwater moni-
toring wells and a temporal sampling frequency (Loaiciga et al., 1992). Since there is a dualism between monitoring costs and monitoring quality (i.e., the information gained by monitoring), economic considerations have the most significant influence on the number of monitoring wells. Considering these two components as axes in a two-dimensional coordinate system, optimal GMNs lie along a Pareto front on which the maximum information content is achieved for the respective budget. Moreover, existing GMNs are usually grown historically regarding the locations and number of monitoring wells and are therefore pri-
marily inefficient. This means that the monitoring quality is relatively low for the given costs. Thus, optimization could reduce the operating costs without loss of monitoring quality by optimizing the monitoring network regarding the number of wells and their location (Emmert et al., 2016). Furthermore, directives such as the European Water Framework Directive (EC, 2000) or the European Nitrate Directive (EC, 1991) demand the integration of regional monitoring networks into national or international networks. Selecting a reasonable subset of these networks capable of capturing the dynamics of the groundwater body
is an essential and challenging task.

Usually, GMNs are classified according to their purpose into groundwater quality monitoring network (GQMN), mostly multivariate, and groundwater quantity/groundwater level monitoring network (GLMN), uni-variate. This classification does not preclude a GMN from performing both tasks. However, optimization approaches usually address one of the two tasks. To date, there are neither standard regulations for the planning and expansion of existing GMNs nor established methods. Instead,
a high degree of subjectivity prevails. In the last decades, many studies have been published dealing with the optimization of GMNs. The widely varying requirements for optimizing monitoring networks led to various approaches that attempt to meet these requirements differently. The choice of method usually depends on the GMN type (GQMN vs. GLMN), scale (local-, regional-, national), uni- or multivariate network, optimization strategy (extension of GMN vs. reduction of redundant wells), and consideration of dynamics (spatial vs. spatio-temporal) and lastly, the purpose of the monitoring network. With the latter,
a distinction is usually made between risk-oriented monitoring (mainly concerning groundwater quality in the catchment of waterworks) and surveillance monitoring, e.g., according to the European Water Framework Directive.

In general, the design of a monitoring network is considered a nonlinear and non-convex optimization problem whose optimally criterion measures the useful information contained in the information matrix of the design (Ushijima et al., 2021). Groundwater monitoring network optimization approaches are commonly divided into three categories based on the techniques
applied: (a) based on hydrogeological conceptual models and hydrogeological expert knowledge, (b) based on numerical groundwater flow models (Kim and Lee, 2007; Singh and Datta, 2016; Thakur, 2017; Sreekanth et al., 2017), and (c) based on data analysis with (geo-)statistical techniques. Many studies have focused on the geostatistical ability of kriging frameworks to determine new monitoring wells based on the reduction of estimation variance as optimization criterion (Nunes et al., 2004; Li et al., 2011; Varouchakis and Hristopulos, 2013; Bhat et al., 2015; Thakur, 2015; Ohmer et al., 2019). With the steady increase





in computational capacity in recent years, there are a growing number of studies that tackle these optimization problems using traditional data-driven heuristic optimization criteria such as genetic algorithm (GA) (Dhar and Patil, 2012; Reed and Kollat, 2013; Khader and McKee, 2014; Puri et al., 2017; Pourshahabi et al., 2018; Ayvaz and Elçi, 2018; Yudina et al., 2021; Komasi and Goudarzi, 2021), artificial neural networks (ANN) (Alizadeh et al., 2018), particle swarm optimizations (Gaur et al., 2013; Guneshwor et al., 2018; De Jesus et al., 2021), support vector machines (SVM), (Asefa et al., 2004; Bashi-Azghadi and

Kerachian, 2010) and relevance vector machines (RVM) (Khalil et al., 2005; Ammar et al., 2008) or a combination of these approaches. Further studies use entropy and information theory based approaches (Hosseini and Kerachian, 2017; Keum et al., 2017; Alizadeh and Mahjouri, 2017), as well as Kalman filtering (KF) (Kollat et al., 2011; Júnez-Ferreira et al., 2016).

In most of the mentioned studies, the optimization of the GMN amounts to a computationally intensive combinatorial search with innumerable multi-dimensional iteration steps since many complex physical systems are described by a high-dimensional

state $[x \in \mathbb{R}^n]$. Moreover, improved data recording and increasing storage leads to fast and strongly growing system complexities and, therefore,to increased computing time beyond Moore's law (Moore, 1965). However, the dynamics of such complex systems often evolve on a low-dimensional attractor, which can be used to predict and control these systems. Pattern extraction is associated with the search for coordinate transformations that simplify the system dynamic and the computational effort (Brunton and Kutz, 2017). In recent years, powerful new techniques in data science have been developed that are capable

of analyzing complex data and extracting essential features and correlations from high-dimensional dynamic systems. Sparse sampling (Candes et al., 2005; Candes and Wakin, 2008; Baraniuk, 2007), sparse reconstruction (Yildirim et al., 2009; Annoni et al., 2018; Castillo and Messina, 2020), and sparse classification (Brunton and Kutz, 2017), enable the recovery of relevant information from remarkably few measurements. Although sparse sampling, such as compressed sensing, is a common and powerful method, often used in other fields of science including seismic and medical image processing, fluid dynamics or

remote sensing, to our knowledge there are only a few studies in the field of hydrogeology that applied sparse sensing for hydrogeological tasks. Hussain and Muhammad (2013) utilized sparse signal extraction methods based on $l_1$ norm minimization to exploit the spatial sparsity in hydrodynamic models and thereby reduce the number of measurements need to reconstruct the signal. Lee et al. (2021) used compressed sensing for generating groundwater level (GWL) contour maps based on sparsely sampled or incomplete data from a groundwater model below the Nyquist-Shannon sampling criterion (Shannon, 1949). They

found that compressed sensing performed much better compared to traditional interpolation methods such as kriging. Ushijima et al. (2021) developed an experimental design algorithm to select locations for a network of monitoring wells with maximum information. The combinatorial search was performed with a GA, combined with a proper orthogonal decomposition (POD) to reduce the computational cost of using the GA. Proper orthogonal decomposition, which is often formulated using the singular value decomposition (SVD)), is a dimensionality reduction method that extracts relevant large coherent structures/patters (low-

dimensional features) from high-dimensional data (Pollard et al., 2017).

This study focuses on a data-driven algorithm to optimize a GMN regarding the number and locations of monitoring wells for temporal and spatial GWL reconstruction. The algorithm uses data-driven sparse sensing techniques and a QR-based sensor placement algorithm, that ranks monitoring wells according their information content. It is based on work by de Silva et al. (2021), Manohar et al. (2018) and Clark et al. (2019), implemented in the *PySensors* package, and has e.g., been successfully





applied in a similar context of sensor placement for sea surface temperature reconstruction, fluid flow data (Manohar et al., 2018; Clark et al., 2019), and wind flow data (Annoni et al., 2018), as well as for classification tasks in e.g. image recognition or cancer classification by microarrays (Brunton et al., 2016). We have adapted this methodology for the first time for the application to GMNs, as we see the following advantages over existing methods: (i) It can simultaneously take spatial as well as temporal information into account, (ii) it allows the ranking of existing monitoring wells based on their information

content, (iii) based on the ranking, an existing network can be reduced, while the values either of the abandoned wells or a spatially continuous GWL can be reconstructed, (iv) it proposes locations for an extension of a network, that account for the best possible gain in knowledge, and (v), if necessary, allows the application of a cost function for the extension of the network (Clark et al., 2019), either to prefer more suitable locations (e.g., in terms of infrastructure) or to exclude certain areas (like e.g., inaccessible terrains, steep slopes, etc.). We apply the adapted algorithm to a real-world GLMN to demonstrate its suitability for

groundwater monitoring networks in general. The data set used for this purpose consists of weekly GWL monitoring between 1990 and 2015 from 480 monitoring sites in the Upper Rhine Graben's upper alluvial aquifer. In particular, we show how the algorithm can be applied to address the following questions regarding the optimization of an existing network:

– What is the ranking of monitoring wells in an existing network in terms of their information content/reconstruction performance, i.e., in which order should the wells be removed if a network reduction is desired?

– How does the reconstruction/interpolation error develop when a given number of monitoring wells are reduced? How does the error of reducing wells according to information content compare to a random reduction?

– When the goal is network extension, , where should new wells be placed for maximum information gain? How much is the increase in information, i.e., how much will the spatial reconstruction error be reduced?

– How well does a combined reduction/extension (i.e., replacement) of a certain number of wells perform compared to a

straightforward extension?

## 2  METHODOLOGY

### 2.1  Mathematical Background

#### 2.1.1  Compressed sensing

Most multi-dimensional natural signals are compressible (resp. sparsely representable). That means, when the signals are

transformed into a convenient coordinate system (basis), only a limited number of basis modes are active. These basis modes correspond to the large mode amplitudes (Brunton and Kutz, 2017). In data compression, for example, *JPEG* or *MP3* compression, only these values are stored to efficiently reconstruct the input signal with a considerable reduction in data size and little loss of information. A compressible signal $\mathbf{x} \in \mathbb{R}^n$ can be written as a sparse vector $\mathbf{s} \in \mathbb{R}^n$ on a new orthonormal basis of $\Psi \in \mathbb{R}^{n \times n}$ such that:






$$\mathbf{x} = \mathbf{\Psi s} \tag{1}$$

Vector $\mathbf{s}$ is $K$-sparse if it is a linear combination of only $K$ basis vectors (exactly $K$ nonzero elements). The theory of compressed sensing uses this principle as it attempts to infer the sparse representation $\mathbf{s}$ in a known transformed basis system with a very small, low-dimensional (compressed) subsample.

$$\mathbf{y} = \mathbf{Cx} = (\mathbf{C\Psi})\mathbf{s} = \mathbf{\Theta s} \tag{2}$$

where the vector $\mathbf{y} \in \mathbb{R}^p$ is a set of incoherent observations and $\mathbf{C} \in \mathbb{R}^{p \times n}$ a observation matrix of $p$ linear observations. $\mathbf{\Theta}$ is the condition number. The objective in compressed sensing is to find the $l_1$-norm of sparsest vector $\hat{\mathbf{s}}$ (under a set of conditions) that is consistent with $\mathbf{y}$:

$$\mathbf{s} = \arg\min_{\mathbf{s}'} \|\mathbf{s}'\|_1 \quad \text{such that} \quad \mathbf{y} = \mathbf{\Theta s}' \tag{3}$$

which almost certainly end up with the sparsest possible solution for $\mathbf{s}$ (Candes et al., 2005; Candes and Wakin, 2008; 125 Donoho, 2006; Baraniuk, 2007).

### 2.1.2 Sparse sensor placement

While compressed sensing uses random measurements to reconstruct high-dimensional unknown data from a universal basis $\mathbf{\Psi} \in \mathbb{R}^{n \times n}$, data-driven sparse sensor placement collects available information about a signal from observed samples to build up a tailored basis $\mathbf{\Psi}_r \in \mathbb{R}^{n \times r}$ for the respective signal and thus to identify optimal sensor placements for the reconstruction of 130 this signal with low-losses. Let the full signal be an unknown linear combination of basis coefficients $\mathbf{a} \in \mathbb{R}^r$ (vector of mode amplitudes of $\mathbf{x}$ in basis $\Psi$):

$$\mathbf{x} = \sum_{k=1}^{r} \psi_k a_k = \mathbf{\Psi}_r \mathbf{a} \tag{4}$$

The central challenge is to design a incoherent (i.e. rows of $\mathbf{C}$ not correlated with columns $\psi$ of $\mathbf{\Psi}_r$) measurement matrix $\mathbf{C}$ that allows to identify the optimal $p$ observations $\mathbf{y}_i$ to accurately reproduce the signal $\mathbf{x}$


$$\mathbf{y} = \mathbf{Cx} = (\mathbf{C\Psi}_r)\mathbf{a} = \mathbf{\Theta a} \tag{5}$$

For $n$ sensor observations and a given $p$ sensor budget, the sampling matrix $\mathbf{C}$ must be structured as follows:

$$\mathbf{C} = \begin{bmatrix} \mathbf{e}_{\gamma 1} & \mathbf{e}_{\gamma 2} & \dots & \mathbf{e}_{\gamma p} \end{bmatrix}^T \tag{6}$$

Here $e_j \in \mathbb{R}^n$ are the canonical basis vectors with unit entry at index $j$ and elsewhere zeros. Thus each row of $\mathbf{C}$ only observes from a single spatial location, corresponding to the sensor location. The observations are made up of $p$ elements 140 selected from $\mathbf{x}$

$$\mathbf{y} = \mathbf{Cx} = \begin{bmatrix} x_{\gamma 1} & x_{\gamma 2} & \dots & x_{\gamma p} \end{bmatrix}^T \tag{7}$$





with $\gamma \in \mathbb{N}^p$ as index set of the sensor locations designates the index with cardinality $|\gamma| = p$ and additionally number of sensors $n \geq r$ of $\Psi$ for a well defined linear inverse problem (Manohar et al., 2018). The unknown $\mathbf{x}$ can thus be reconstructed by approximating $\mathbf{a}$ with the Moore-Penrose pseudoinverse of (5) to:

$$\mathbf{C}^{\star} = \underset{\mathbf{C} \in \mathbb{R}^{p \times n}}{\arg\min} |\mathbf{x} - \mathbf{\Psi}(\mathbf{C}\mathbf{\Psi})^{\dagger} \mathbf{y}|_2^2 \tag{8}$$

Where $\dagger$ denotes the Moore-Penrose pseudoinverse, it is assumed that optimal sensor selection $\mathbf{C}^{\star}$ is a mostly sparse subset selection operator, the nonzero entries in the rows represents the monitoring wells.

### 2.1.3 Taylored basis $\mathbf{\Psi}_r$

As described above, in data-driven sparse sensing the universal basis $\mathbf{\Psi}$ is replaced by a tailored basis $\mathbf{\Psi}_r$, which is built from the training data $\mathbf{X}^{tr}$, e.g., by using dimensionality reduction techniques. In this study, we are using three basis types which are typically used for sparse sensor selection:

**Identity basis**: Centered raw data is used directly without dimensionality reduction. $\mathbf{\Psi}_r = \mathbf{X}^{tr}$. Since no low-rank approximation of the data is performed, no information is lost. However, this comes at the cost of a longer computation time (de Silva et al., 2021).

**Random Projection basis**: Dimensionality is reduced by projecting the input data onto a randomly generated matrix $\mathbf{\Psi}^r = \mathbf{G}\mathbf{X}^{tr}$ where the entries $\mathbf{G} \in \mathbb{R}^{2p \times m}$ are drawn from a Gaussian density function with mean zero and variance $1/n$ (Dasgupta, 2000; Li et al., 2006).

**SVD/principal component analysis (PCA)**: Linear dimensionality reduction is performed using a truncated SVD. Singular value decomposition is a numerically robust and efficient method for extracting dominant patterns from low-dimension (Golub and Kahan, 1965; Halko et al., 2011). For a matrix $\mathbf{X} \in \mathbb{C}^{n \times n}$, the SVD is given by:

$$\mathbf{X} = \mathbf{U}\mathbf{\Sigma}\mathbf{V}^T = \mathbf{\Psi}\mathbf{\Sigma}\mathbf{V}^T \approx \mathbf{\Psi}_r\mathbf{\Sigma}_r\mathbf{V}_r^T \qquad \text{where} \qquad \mathbf{U} \in \mathbb{R}^{n \times r}, \mathbf{\Sigma} \in \mathbb{R}^{r \times r}, \mathbf{V} \in \mathbb{R}^{m \times r}, \tag{9}$$

The columns of $\mathbf{\Psi}$ are the left singular vector of $\mathbf{X}$. They are often termed as spatial correlations, principal components, features or POD modes of the data set. $\mathbf{\Sigma}$ is diagonal matrix.

### 2.1.4 QR Pivoting for sparse sensors

While the previous steps serve to best fit the basis to the training data, the next steps aim to determine the resulting optimal sensor locations that minimize the reconstruction error. This optimization problem is solved using an approximate greedy solution using reduced QR factorization with column pivoting (Brunton and Kutz, 2017; Halko et al., 2011). QR factorization decomposes a matrix $\mathbf{A} \in \mathbb{R}^{m \times n}$ into a unitary matrix $\mathbf{Q}$, and an upper-triangular matrix $\mathbf{R}$ as well as a column permutation matrix, in our case $\mathbf{C}$ (6) such that $\mathbf{A}\mathbf{C}^T = \mathbf{Q}\mathbf{R}$. QR pivoting increments the volume of the submatrix constructed from the pivoted columns by selecting a new pivot column with a maximum 2-norm within all modes in the library and then subtracting from the other column its orthogonal projection onto the pivot column. Thus, QR factorization with column pivoting yields $r$ column indices (which corresponds to sensor locations) that best sample the $r$ basis modes (columns) $\mathbf{\Psi}_r^T$ .



$$\mathbf{\Psi_r}^T \mathbf{C}^T = \mathbf{QR} \tag{10}$$

Since the pivots columns represent the sensors, the QR-factorization results in a hierarchical list of all $n$ pivots, where the first $p$ pivots are optimized for the reconstruction of $\mathbf{\Psi}_r$. This means that in the GMN optimization based on hydrograph data, all monitoring wells are ranked based on their information content. When using spatial input data, e.g., from interpolation or model results, all gridded input data cells are ranked based on their information content. Thus it allows recommendations for the placement of additional monitoring wells at locations with supposedly high information content. The used QR decomposition approach includes a cost constraint function (Clark et al., 2019). This constraint allows different "costs" to be considered when selecting sensor placement, such as favoring or excluding certain areas.

## 2.2 Application Cases

In principle, there are two possible application cases of the algorithm regarding groundwater monitoring data: (i) the application to the observed data at the wells (i.e., hydrographs) only and (ii) the application of spatially continuous gridded information that has been regionalized based on the well data (e.g., by interpolation). While the optimization based on hydrographs only serves to rank the individual wells of the network according to their information content and thus to identify and eliminate redundant wells, the spatially continuous input data also allow a GMN extension at optimal locations, as well as the best possible reduction of the spatial prediction error.

## 2.3 Error metrics

The calculation of the reconstruction error for a given set of measurements is done using the root mean square error (RMSE) for the scoring function. We further used the following metrics widely used for calibration and evaluation of hydrological models: mean absolute error (MAE), Nash-Sutcliffe efficiency (NSE), Kling-Gupta efficiency (KGE), squared Pearson's correlation coefficient ($R^2$) and relative Bias (rBias). In the following equations, $o$ stands for observed values, $r$ for the reconstructed values, $cov$ is the covariance, $\sigma$ is the standard deviation, $\mu$ is the arithmetic mean, $n$ stands for the number of measurements.

The RMSE is one of the most commonly used error-index statistics. In general, the lower the RMSE, the better the model performance.

$$RMSE = \sqrt{\frac{1}{n}\sum_{i=1}^{n}[o_i - r_i]^2} \tag{11}$$

Analogous to the RMSE, the smaller the MAE, the better the performance

$$MAE = \frac{1}{n}\sum_{i=1}^{n}|o_i - r_i| \tag{12}$$

The NSE (Nash and Sutcliffe, 1970) is a widely used goodness of fit measure of hydrologic models as it normalises model performance into an interpretable scale (Knoben et al., 2019). The NSE ranges between -∞ and 1, where 1 indicates a perfect correspondence between observations and reconstructions, while a NSE = 0 indicates that the model has the same explanatory power as $\mu(o)$.



$$NSE = 1 - \frac{\sum_{i=1}^{n}[o_i - r_i]^2}{\sum_{i=1}^{n}[o_i - \mu(o)]^2} \tag{13}$$

The KGE (Gupta et al., 2009) was proposed as an alternative to the NSE because it addresses several shortcomings of the
NSE (Knoben et al., 2019). Like the NSE, a KGE = 1 indicates a perfect model correspondence. However, explicit statements
on benchmark performance have so far varied .

$$KGE = 1 - \sqrt{[r-1]^2 + [\alpha-1]^2 + [\beta-1]^2} \quad \text{with:} \quad r = \frac{cov(o,r)}{\sigma(o)\sigma(r)}, \quad \alpha = \frac{\sigma(r)}{\sigma(o)}, \quad \beta = \frac{\mu(r)}{\mu(o)} \tag{14}$$

where $r$ is the linear correlation between $o$ and $r$, $\alpha$ is a measure of the variability error, and $\beta$ is a bias term.

We use the squared Pearson ($r$, eq. 14 ) correlation coefficient as a general coefficient $R^2$. It describes the degree of collinear-
ity between measured and reconstructed data. $R^2$ ranges from 0 to 1, with higher values indicating lower error variance. In
general, values above 0.5 are considered acceptable.

$$R^2 = \left[\frac{cov(o,r)}{\sigma(o)\sigma(r)}\right]^2 \tag{15}$$

The relative bias is a measure for a systematic over- or underestimation of a model. The optimal rBias is 0. Positive values
indicate model underestimation bias; negative values indicate model overestimation bias (Gupta et al., 1999).

$$rBias = \frac{1}{n}\sum_{i=1}^{n}[\frac{o_i - r_i}{o_{max} - o_{min}}] \tag{16}$$

Statements about model performance in the Results and Discussion section are based on Moriasi et al. (2007) guidelines for
model evaluation.

## 2.4 Data and Study Area

### 2.4.1 Hydrogeological Framework

The Upper Rhine Graben (URG), also known as Rhine Rift Valley, is a 300 km long and, on average, 50 km wide structural
trough. It was formed in the Oligocene in response to the alpine orogeneses and subsequently filled with fluvial to lacustrine
sediments of Late Miocene, Pliocene, and Quaternary (Przyrowski and Schäfer, 2015). The Pliocene and Quaternary alluvial
gravels and sands represent the largest groundwater reservoir in central Europe (LUBW, 2006). Based on their permeability
and the appearance of fine-grained horizons, the Pliocene and Quaternary gravels are subdivided into three (locally also more)
aquifers, partly separated by fine-grained sediments (Wirsing and Luz, 2007). The study region is in the Baden-Wuerttemberg
part of the URG (Fig. 1). The Rhine forms the western boundary, Kaiserstuhl volcano complex is the southern boundary.
To the East, the URG is bounded by a rift flank uplift composed of a system of troughs and highs, which follow the ENE
structural grain of the Variscan fold belt (Derer, 2003). Along the study area these are, from South to North, the Black Forest
High, Kraichgau-Basin, and Odenwald-Spessart-High. Groundwater recharge occurs predominantly through lateral inflow and
infiltration of streams from the Black Forest valleys in the East, the Freiburger Basin in the Southwest, and the infiltration of
the Rhine and other surface waters.

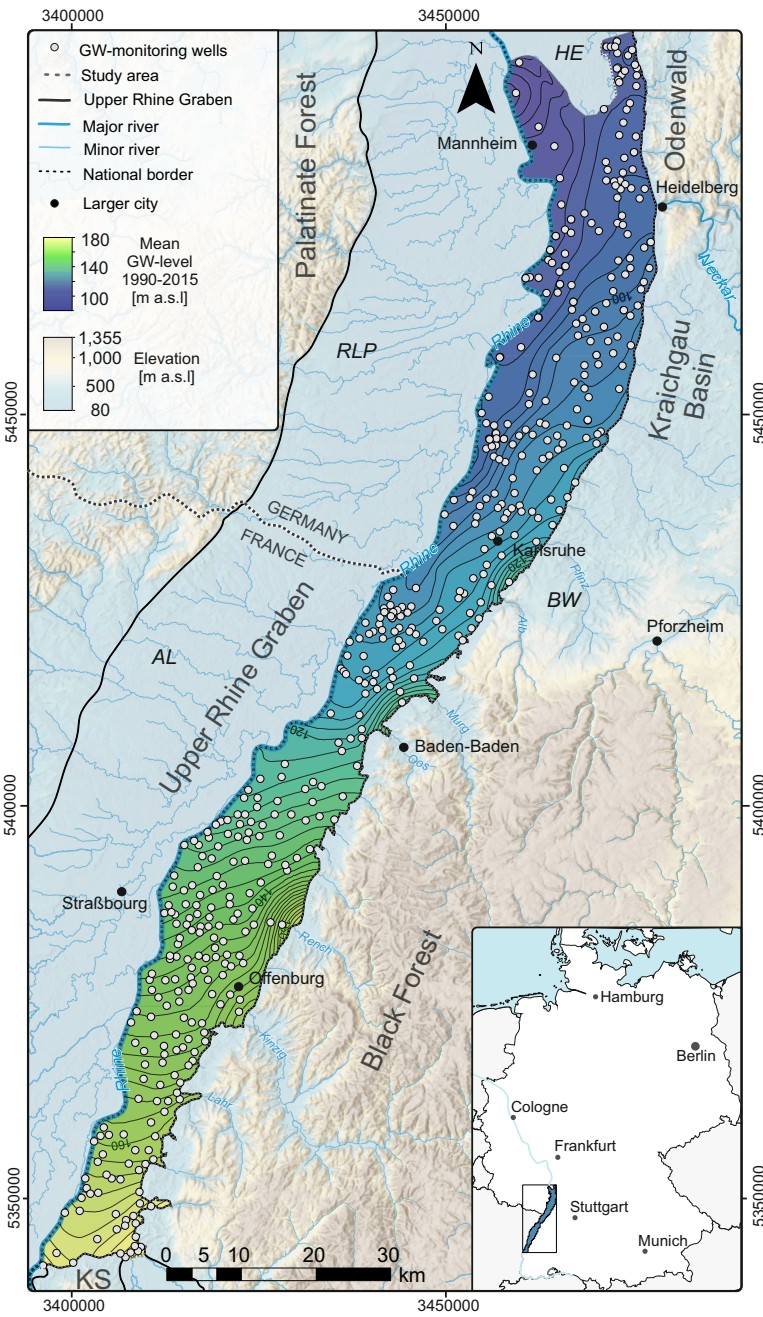

**Figure 1.** Study area within the URG and the 480 groundwater monitoring wells used for optimization. RP: Rhineland-Palatinate; He: Hesse, BW: Baden-Wuerttemberg, Al: Alsace (France); KS: Kaiserstuhl volcanic complex.



### 2.4.2 Data and Preprocessing

The data set used in this study consists of weekly GWL measurements from 480 wells in the uppermost aquifer within the Quaternary sand/gravel deposits of the URG, covering the period from 1990 to 2015 (i.e., 1304 time-steps). Outlier values that
exceeded a moving average (window size 11) of $\pm 3\,\sigma$ were removed during preprocessing. Data gaps were subsequently filled based on information from highly correlated neighboring hydrographs using the clustering results of Wunsch et al. (2021). Where this did not yield plausible results or was not possible due to similarly missing data in neighboring hydrographs, an alternative PCHIP (piecewise cubic hermite interpolating polynomial) interpolation was performed. To avoid possible bias, the measurement data are globally and locally centered. The data set is split into two subsets: the first 80% (01/1990-12/2009,
1043 time-steps) is used to train the algorithm, the last 20% for test/validation (01/2009-12/2014, 261 time-steps). The well data are publicly available at the Baden-Wuerttemberg State Office for Environment web service (LUBW, 2021).

### 2.4.3 Groundwater contour maps

We used the hydrograph data to generate 1304 weekly groundwater contour maps with a grid size of $50 \times 50$ m using ordinary kriging. In addition to finding a sparse set of monitoring wells for the optimal temporal reconstruction of other hydrographs,
the objective is to identify monitoring wells that allow optimal spatio-temporal reconstruction of GWLs from a subset of the wells. Moreover, the spatially continuous information of the gridded contour maps is used to suggest additional locations for an extended network. We used an omnidirectional Gaussian semivariogram model for interpolation, which is flexible and, therefore, a good candidate for a standard model (Krivoruchko, 2011). The associated parameters partial sill, range, search neighborhood, and specific search distance were optimized using automatic cross-validation (CV) diagnostics, minimizing the
MAE for each individual case ($\text{MAE}_{mean} = 0.57$ m; $\text{MAE}_{min} = 0.04$ m; $\text{MAE}_{max} = 4.04$ m). It should be noted that the use of a single variogram model (Gaussian) may not be the optimal way to quantify spatial correlation, especially for nonstationary data. However, this is a necessary simplification due to the automation process that still produces comparable interpolation results, while the best possible interpolation result is not the focus of this study. Just as with the hydrographs, the first 80% (01/1990-12/2009, 1043 time-steps) of the contour maps were used to train the algorithm, the last 20% for test/validation
(01/2009-12/2014, 261 time-steps).

## 3 RESULTS AND DISCUSSION

The following section is structured as follows. First, the grid search results regarding the three types of basis used, the number of basis modes, and a varying number of sensors are presented and discussed. This is followed by the results of the GMN optimization-based solely on the hydrograph data set. Finally, the results of the GMN optimization with the interpolated GWL
as inputs are shown.



## 3.1 Grid Search Results

Fig. 2a shows the RMSE between the estimated and actual GWLs for the validation set as a function of the number of wells (sensors), that the network is reduced to, the type of tailored basis (identity basis, random projection, and SVD basis), and the number of basis modes used. Since the number of sensors and basis modes interacts, it is necessary to determine the appropriate

number of basis modes that will result in the lowest reconstruction error for a given number of sensors. If the number of sensors is close to the number of basis modes, the reconstruction error increases significantly for all three basis modes types used. While the number of basis modes for identity and random projection is theoretically open-ended, the dimensionality for SVD, thus the number of basis modes, must be less than the number of sensors. Although there are SVD-methods (e.g., randomized SVD, (Halko et al., 2011)) that allow oversampling with an additional number of random vectors, our results show that the accuracy

of SVD generally decreases as the number of basis modes increases, which is consistent with the findings of de Silva et al. (2021). Therefore, we decided against an SVD method with oversampling and opted for the truncated SVD implemented in *PySensors* (de Silva et al., 2021), thus using a maximum of 480 basis modes. All 1043 time-steps (in the training set) were used as the maximum number of basis modes for identity and random projection. According to previous studies, the number of basis modes should be at least equal to the number of sensors $p + 10$ (de Silva et al., 2021). Clark et al. (2019) used $2p$ basis

modes, which in our case equals a maximum of 960 (for all 480 sensors), and thus is covered by the maximum of 1043 basis modes in the grid search.

The results show that with only a few basis modes, SVD has the highest accuracy (Fig. 2a). As the number of basis modes exceeds the number of remaining sensors, the identity and random projection basis perform better. In general, random projection and identity basis perform similarly. With less than 950 basis modes, slightly better results are obtained with random projec-

tion; above 950 identity basis performs marginally better. Fig. 2b shows the lowest RMSE per number of sensors achieved in the grid search, Fig. 2c shows the corresponding number of basis modes. The gray dashed line in Fig. 2b shows the median of reconstruction errors from 100 iterations with a random sensor selection as a benchmark. Except for SVD basis with less than 50 sensors, all three basis types perform considerably better compared to reconstruction with the randomly placed sensors, independently of the number of sensors. Our findings are consistent with those of Manohar et al. (2018); Clark et al. (2019),

and de Silva et al. (2021) where SVD consistently underperforms compared to random projection and identity basis. However, the latter two show almost identical results with an optimized number of basis modes (Fig. 2b), with random projection performing marginally better with a small number of sensors and identity basis performing slightly better with a larger number of sensors. For consistency reasons and based on the grid search results, we decided to compute the optimization steps shown in the following uniformly with identity basis and a fixed number of 1043 basis modes. This combination, on average, shows the

best results for any chosen number of remaining sensors (Fig. 2c).

As examples, all results of the following section are shown using five reduction stages, to 90%, 75%, 50%, 25%, and 10% of the original number of monitoring wells/sensors. Thus, e.g., at the 75% stage, the most optimal 75% of the monitoring wells are selected to keep and used to predict the time series/hydrographs of the removed 25% wells. The reduction stages are also shown in the grid search results (Fig. 2b and 2c). The color scheme of the reduction stages is kept for all remaining figures.





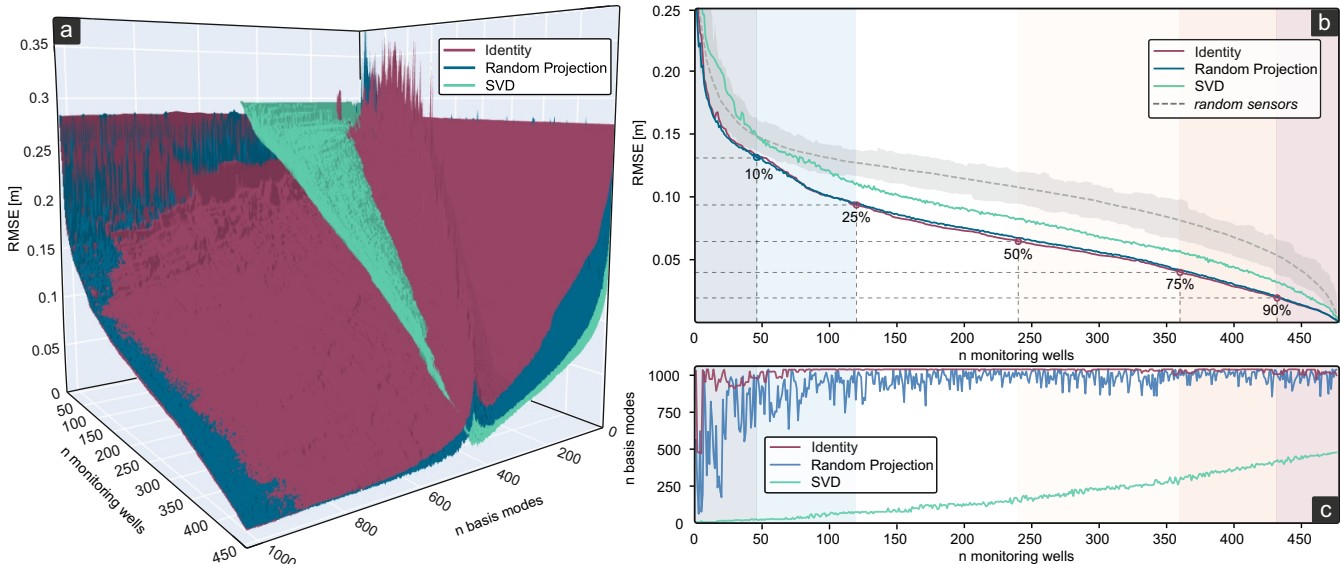

**Figure 2.** Grid search CV results. (a) Reconstruction error (RMSE) vs. the number of monitoring wells and the number of basis modes for the three basis projections described in basis: identity ($\mathbf{\Psi}_r = \mathbf{X}^{tr}$), random projection ($\mathbf{\Psi}_r = \mathbf{G}\mathbf{X}^{tr}$), SVD ($\mathbf{\Psi}_r = \mathbf{U}^{tr}$). (b) Minimum RMSE achieved with a given number of QR-selected monitoring wells (Pareto front) and the same number of randomly selected wells (median of 100 runs, grey shading represents the total range) for comparison. (c) The corresponding number of basis modes, that leads to the lowest RMSE in b. Accordingly, b and c represent a section through a at the respective minimum on the axis n monitoring wells.

## 3.2 Ranking of wells and network reduction with hydrograph data

Fig. 3 shows the result of the ranking of the monitoring wells, computed with an identity basis and using 1043 basis modes. The ranks are assigned from 1 (essential well with high information content) to 480 (most redundant well); thus, lower numbers mean a higher ranking concerning their information content and importance to reconstructing potentially removed redundant wells.

The color scheme ranges from dark red for redundant monitoring wells that can be eliminated with a minor loss of prediction accuracy to dark blue for important monitoring wells that contain essential information about the system and are needed for the accurate reconstruction of signals at other monitoring wells. In addition, Fig. 4 shows the centered hydrographs of the most important 10% and the most redundant 10% of all hydrographs. Most of the important wells (blue, <10%) show a pronounced flashiness (i.e., high frequency and rapidity of short-term changes) and strong irregular patterns during the recording period. These dynamics indicate strong interaction with surface waters or boundary inflows, for example, from side valleys of the rift flanks, which can also be seen in Fig. 3 from the location of the wells. Additionally, the most important wells include those with a distinct trend, which can be best seen for the two highest-ranked wells at the bottom, showing an upward trend over the considered period.

**Figure 3.** QR based ranking of groundwater monitoring wells based on hydrograph information content, from 1 (high information content, blue) to 480 (redundant, red). The hydrographs of the highlighted wells (pink) are shown in Fig. 6.

**Figure 4.** Stacked z-transformed hydrographs of the 10% "most important" monitoring wells (blue, bottom) and the 10% "most redundant" monitoring wells (red, top). Coloring and stacked order reflecting ranking order.





In contrast, the redundant wells show low flashiness and also include wells with high seasonality, though most of the signals
seem to be dominated by inter-annual variations. Most of these wells are located in the northern part of the study area within
the URG. Since the eastern boundary in this area is the Kraichgau Basin, the landscape profile is less pronounced than in the
Black Forest hill range in the south and the Odenwald in the north. Therefore, less recharge occurs through stream infiltration,
which is often the reason for more pronounced short-term variations or flashiness. The hydrographs of the wells <90% to >10%
can be found in the electronic appendix. Overall, the ranking exhibits that the most important wells include the ones with an
noticeably unusual behavior, i.e., patterns that are not present in many of the other wells (like flashiness, trends, and jumps) and
thus are hard to reconstruct. Overall, the more redundant wells either show a higher seasonality or tend to show low variability.
Both patterns are common to a larger number of the wells and can be reconstructed more easily.

While the ranking itself contains already essential information and could be used, for example, to equip higher-ranked
monitoring wells with higher quality sensors or measure them with a higher time-frequency, we use the ranks here to reduce
the original network well by well, with most of the results shown only for the five above mentioned reduction stages, to 90%,
75%, 50%, 25%, and 10% of the original number of monitoring wells.

The upper part of Fig. 5 shows the development of the prediction accuracy of the GMN reduction for the error measures NSE,
KGE, and $R^2$ (left) as well as rBias, RMSE, and MAE (right) for the validation data set (mean and ranges of the reconstructed
validation period of all predicted/removed wells). Even with only a few optimally selected wells, the predictive power is
considerably higher than the mean value of the time series (NSE = 0). An average performance for the validation period of
all predicted removed wells rated "satisfactory" (NSE > 0.5) is already achieved with only nine remaining wells (1.9%), rated
as "good" (NSE > 0.65) with 21 remaining wells (4.4%) and "very good" (NSE > 0.75) with 54 remaining wells (11.3%).
With more than 191 wells (39.8%), the NSE rises above 0.9. KGE and $R^2$ behave in much the same way as NSE. A KGE of
0.75 is achieved with nine wells (1.9%), and 0.9 with 144 wells (30.0%). $R^2$ of 0.75 is achieved with 22 wells (4.6%), and
of 0.9 with 155 wells (32.3%), respectively. A mean MAE of 0.1 m is achieved with 31 monitoring wells (6.5%) remaining,
0.05 m with 147 wells (30.6%), and 0.01 m with 394 wells (82.1%). Below about 25% of sensors, the removal of each sensor
leads to a disproportionately decrease in accuracy, with a very steep drop from below about 5%. From about 25% upwards, the
gradient gradually becomes more linear, meaning a linear (but small) performance increase with more monitoring wells. The
rBias approaches zero also at about 25% remaining wells. Thus, we conclude that about 25% of the wells could be seen as a
kind of absolute minimum that is required to adequately describe the system dynamics in the considered study area, despite
the average NSE of the reconstruction still being rated as "good" for only about the optimally selected 10% of the wells.

The lower part of Fig. 5 displays the performance of the QR-optimized sensors compared to an equal number of randomly
selected remaining wells at the five reduction stages. Removing wells based on the ranking results leads to less prediction loss
for all stages except 10%, where the average errors are about the same. Again, from about 25% upwards, the performance
differences get more pronounced, with a considerably higher average, 25%-quantile, and minimum NSE, KGE, and $R^2$ values
(lower MAE and RMSE, respectively). From about 75% upwards, also the 75%-quantile and minimum NSE, KGE, and $R^2$
values are clearly higher (lower for MAE and RMSE, respectively). This clearly shows that the advantages of the data-driven
optimization method come into play, especially for moderate to smaller reductions of a GMN.



**Figure 5.** Top: Reconstruction error metrics (KGE, R$^2$, NSE, MAE, RMSE, rBias) as a function of the number of QR-ranked remaining monitoring wells for the identity basis and 1043 basis modes (lines are mean values, shading represents total range of recontstruction errors over all removed wells). Bottom: Same reconstruction error metrics at the reductions steps 10%, 25%, 50%, 75%, and 90%, as boxplots over all removed wells, and compared to the same number of randomly removed monitoring wells.

Fig. 6 shows the temporal reconstruction accuracy achieved at the considered reduction stages from 10% to 90% for eight

selected wells (see also Fig. 3). These stages were chosen to reflect the dynamics spectrum and represent the full ranking range. The reconstruction is always based on the higher-ranked remaining wells (but keeping the chosen reduction stages). Consequently, well 154-304-1, the highest-ranked well shown with rank 59 (bottom), which could theoretically be reconstructed with a maximum of 58 remaining wells, is reconstructed with 10% of the wells (48). Similarly, well 132-257-4, the lowest-ranked well with rank 478 (top), which could theoretically be reconstructed with a maximum of 477 wells, is reconstructed with 90%,

75%, 50%, 25%, and 10% remaining wells (432, 360, 240, 120 and 48, respectively) for a comparison.



**Figure 6.** Hydrograph reconstruction at the five reduction stages 10% - 90% exemplary shown for eight monitoring wells, along with the respective error measures.





The results show that the individual dynamics of the hydrographs are already adequately reconstructed with a 10% subset of the monitoring network. As expected, as the number of wells increases, the accuracy on average improves, hydrographs are reproduced more consistently, and short-term peaks are reproduced more accurately. Though these seem only comparatively slight improvements considering the overall dynamics, for some wells and time-steps, the absolute errors can be up to sev-
eral tens of centimeters, albeit achieved by many additional wells. Whether this justifies the increased operating costs of the monitoring network depends on the task at hand. The reconstruction results for the other wells can be found in the electronic appendix.

### 3.3 Network reduction and extension based on gridded GWL contour maps

This application case is based on spatially continuous gridded weekly GWL contour maps from 1990 to 2015. Analogous to
the hydrograph data set, the first 80% of this period was used for model training, the last 20% for evaluation. According to the ranking, we investigate how well the GWL can be reconstructed with two reduction stages in which 10% and 20% of the GMN are removed. We have selected these reduction stages since in the analysis of the reduction stages with hydrographs, the advantages of the data-driven optimization method were more pronounced for moderate to smaller reductions of a GMN. Moreover, reducing an existing network to less than 80% remaining wells seems unrealistic in practice. Furthermore, we extend
the existing network by 10% and 20% wells and analyze where new wells are placed, to supposedly improve the GMN. To account for more or less suitable locations (e.g., with regard to infrastructure), we apply "costs" with a non-uniform spatial step function on a $50 \times 50$ grid (corresponding to the used GWL grid) into the QR factorization. The cost function grid is assigned zero (no additional costs) at existing monitoring well locations to ensure that existing wells remain since, technically, the extension (by e.g., 10%) is realized such that 110% of "new" wells are placed on the gridded GWL maps. At potentially
well-suited additional sites, which are defined within 50 m of roads and paths, outside surface waters, and where the slope is less than 20%, we assigned a cost value of 21. For all other areas that are considered as not suitable, the cost weighting is set to 22. Alternatively, a gradual cost function can be used, where the weighting increases with distance to the infrastructure or similar. It should be noted that the weighting depends on the system, basis, and cost function and must be adjusted for the particular case (Clark et al., 2019). We assigned the mentioned weighting factors iteratively until it resulted in the desired
behavior. With the weightings chosen in this way, it was possible to achieve that the first 480 sensors are placed at existing monitoring wells, and all subsequent sensors are placed at suitable locations, while the algorithm avoids the other locations. Finally, we combine a reduction/extension scenario, where the original number of sensors is kept, but the 10% and 20% most redundant sensors are removed and replaced afterward. Technically, this is done in a two-step procedure, consisting of the reduction step, followed by the above described extension, where the cost function is adapted for the existing wells.
The maps on the left side of Fig. 7a show the spatial distribution of the monitoring wells as a result of a 10% (yellow dots) and 20% (red dots) reduction (1), and a 10% (light blue dots) and 20% (dark blue dots) extension (2) of the GMN, as well as a combined reduction/extension of the GMN (3). For the latter, redundant 10% and 20% monitoring wells were eliminated and replaced with wells at optimal locations. We should note that the ranking based on the spatially interpolated data is different from the ranking based on the hydrographs alone (see appendix A1).





**Figure 7.** (a): Location of removed monitoring wells in a 10% (yellow) and 20% (red) QR-based monitoring well reduction (left map, a₁),
in a 10% (light blue) and 20% extension (center map, a₂), and a combined reduction/extension in which 10% and 20%, respectively, of
the monitoring wells were removed and replaced with wells at optimal locations (right map, a₃). (b): Cost function grid used for the GMN
extension. (c): Boxplots show the mean and max. absolute error of the reconstruction of the 216 GWL contour maps of the test set obtained
with the mentioned GMN reduction/extension.





This variation can be explained by the ranking reflecting the information content regarding the reconstruction with the lowest possible error. While in the case of hydrographs, the goal is to reconstruct the hydrographs of the removed wells, here, the goal is to reconstruct the interpolated surface (which constitutes a best guess of spatially continuous GWL based on the available data).

While the first 10% of reduced wells are evenly distributed across the study area, the subsequent removal step (i.e. additional
10%, thus, 20% removed wells in total) eliminates well clusters in the central and northern regions. This seems conclusive because clusters of nearby wells tend to show similar dynamics and thus do not add much information to an interpolation according to Tobler's law. Optimal locations for additional wells are identified primarily along the western and eastern margins, i.e., along the Rhine and downstream of the alluvial valley aquifers of the adjacent Black Forest. These are areas with expected higher groundwater dynamics (e.g., high seasonal magnitudes, high flashiness) and, on the other hand, due to the elongated
geometry of the URG, areas with increased interpolation uncertainty (transition from interpolation to extrapolation). Optimal well locations are primarily, but not exclusively, located in areas of increased variability (standard deviation of the interpolated GWL), 7a (2) and (3).

The boxplots in 7c show the mean (left) and maximum (right) absolute error of the reconstructed 261 GWL contour maps of the evaluation set for all above mentioned scenarios. It has to be noted, that the MAE is now taken as the mean over the spatial
axis, i.e. for each of the reconstructed 261 GWL contour maps separately (whereas with the hydrographs, the MAE was taken as the mean over the time axis for each reconstructed well). This was done, because in the application case, the focus is on the error of a spatial reconstruction of GWL contours, not explicitly on time series. Correspondingly, the maximum absolute error (maxAE) is the maximum over the spatial axis for each of the reconstructed 261 GWL contour maps. We therefore also refer to them as mean and maximum spatial reconstruction errors. Thus, the boxes in 7c show the variability of the mean
absolute error and maximum absolute error over the 261 time-steps.

On average, the model can reconstruct the GWL contour maps with very high accuracy, with mean absolute errors far below 1 cm. This seems very low compared to the reconstruction of the hydrographs. However, this is due to the fact, that the reconstruction of a large number of many similar values (i.e. raster pixels) is much easier for the model, for two reasons: (i) there are much more training patterns for each type of dynamics than with the hydrographs alone, and (ii) the overall
dynamics is reduced by the interpolation itself, which smoothes the data spatially and temporally. Taking these limitations due to spatially interpolated data into account, it seems more reasonable to focus on the maximum absolute error, which allows the identification of areas with higher errors, where the model (with the existing data) can't produce reliable reconstructions, and additional wells would bring the most information.

When comparing the maxAE (7c, right) for all scenarios, we see that a reduction of the network increases the spatial
reconstruction error by a factor of about 2 for 10% reduction and about 3 to 4 for 20%. For comparison, the grey box (100%) shows the reconstruction errors with an unchanged GMN (this error results from the fact that the model is trained with the first 80% of all time-steps, but the reconstruction is performed for the unknown 20% of the evaluation data set). An extension of the network by 10% can considerably reduce the spatial reconstruction error to about less than two-thirds, while an extension by 20% reduces it further to one-tenth of the initial value. Most interestingly, the reconstruction errors for the combined





reduction/extension scenarios with 90%/10% and 80%/20%, respectively, (thus an unchanged number of 480 wells in total), are slightly below the straightforward GMN extension with 110% (528 wells) and 120% (576 wells). To a lesser degree, this also applies to the mean absolute errors, at least for the 80%/20% scenario, which performs slightly better than an extension by 20%, and considerably better than a 10% extension. In practice, that means that with a combined reduction/extension, for example, sensors/data loggers that become available can be used elsewhere at better locations. This reduces the installation

costs of the additional wells and the operating costs of the GMN and moreover performs about equally or even better than a pure extension.

## 4 Conclusions

This study investigated data-driven sparse sensing approaches based on the work of (de Silva et al., 2021; Clark et al., 2019; Manohar et al., 2018) and adapted them to optimize an existing groundwater level monitoring network. The algorithm fits a

tailored basis to the training data, subsequently used in a QR decomposition to rank the monitoring wells by "importance" based on reconstruction performance. This approach allows to remove groundwater monitoring wells with low information content if needed, equip monitoring wells with higher rank with higher quality sensors, or measure with a higher time-frequency. When using spatially continuous input data (by interpolation or numerical simulation), the ranking is performed according to the same scheme for all locations. This rank can be used as a decision-making aid to search for locations for additional monitoring

wells. We incorporated a cost function to eliminate inaccessible locations from the site selection process. Adjusting the cost constraint allows a specific adaptation to the individual problem definition.

Our results show that identifying redundant low-ranking monitoring sites allows a drastic reduction of the monitoring network with minor loss of information, compared to a random reduction (which corresponds to a reduction based on other criteria, as is often the case in practice). In the case of a desired network extension, the reconstruction quality can benefit from

the additional removal of unsuitable wells.

As in related previous studies (Clark et al., 2019; Manohar et al., 2018), using the identity data basis (raw data without dimensionality reduction) and the total number of available base modes yielded a lower reconstruction error for a given number of wells compared to other basis mode types and numbers. This is because no information is lost to construct a low-rank approximation to the data. However, for larger data sets than the one used in this study, an optimization without previous

dimensionality reduction can lead to impractically long computation times. Just as in the work of Clark et al. (2019), and Manohar et al. (2018), a randomized projection of the data in our study performed on average only slightly worse than the raw data and may be worthwhile for large data sets or multiple computational runs due to lower computational costs. Even though the widely used SVD basis gave the worst results for our data set, the reconstruction errors are still lower than for a random network optimization.

In addition to GLMN, this approach can also optimize groundwater quality or multivariate monitoring networks. As with all data-driven methods, the quality of the results depends strongly on the availability of the input data (spatial and primarily temporal). Since this approach relies on detecting patterns in data and placing monitoring locations based on those patterns, it

benefits from large data sets. Therefore, we see the main application of this technique in optimizing monitoring networks at the river basin level according to the Water Framework Directives, where a comprehensive overview of the variability and quantity

of groundwater bodies and the assessment of long-term changes in natural conditions is the monitoring objective.

Overall, we could demonstrate that modern data-driven methods of sparse sensing are well suited for the application to groundwater monitoring networks, as long as there is a good historic data basis. The applied method can be used for an optimization regarding the number of wells and their location, for a network-reduction and extension or both combined. It allows to include a cost-function to account for more or less suitable areas for new wells, and can help to get a maximum

information content for a given budget.

*Code and data availability.* The work is based on the methodology developed by (de Silva et al., 2021; Manohar et al., 2018; Clark et al., 2019) for optimized sensor placement implemented in *Pysensors*, available at: https://github.com/dynamicslab/pysensors.The well data are publicly available at the web service of the Baden-Wuerttemberg State Office for Environment (LUBW, 2021). Our customized code files for the optimization of groundwater monitoring networks are available on: Ohmer (2022)

*Author contributions.* M.O.: Conceptualization, Methodology, Software, Formal analysis, Validation, Investigation, Visualization, Writing – original draft preparation. T.L. Conceptualization, Methodology, Writing – review and editing, Supervision. A.W.: Data preprocessing and curation, Writing – review and editing

*Competing interests.* On behalf of all authors, the corresponding author states that there is no conflict of interest.

*Acknowledgements.* This study is a contribution to the project: Nitrate Monitoring 4.0 - Intelligent Systems for Sustainable Reduction of

Nitrate in Groundwater (NiMo 4.0), funded by the German Federal Ministry for the Environment, Nature Conservation and Nuclear Safety (BMU) on the basis of a resolution of the German Bundestag.



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

**Figure A1.** Comparison of QR-based ranking with 1D hydrograph data (left, $a_1$), with 2D interpolated GWL contour maps (middle, $a_2$), and differences of rankings between 1D and 2D (right, $a_3$).



| | |
|---|---|
| **CV** | cross-validation |
| **GA** | genetic algorithm |
| **GLMN** | groundwater level monitoring network |
| **GMN** | groundwater monitoring network |
| **GQMN** | groundwater quality monitoring network |
| **GWL** | groundwater level |
| **KGE** | Kling-Gupta efficiency |
| **MAE** | mean absolute error |
| **maxAE** | maximum absolute error |
| **NSE** | Nash-Sutcliffe efficiency |
| **R²** | coefficient of determination here: squared Pearson r |
| **PCA** | principal component analysis |
| **POD** | proper orthogonal decomposition |
| **rBias** | relative Bias |
| **RMSE** | root mean square error |
| **SVD** | singular value decomposition |
| **URG** | Upper Rhine Graben |
| **WFD** | Water Framework Directive |
| **ND** | Nitrate Directive |



