# Peer review of "Spatio-temporal optimization of groundwater monitoring networks using data-driven sparse sensing methods"

_Hydrology and Earth System Sciences, 2022_

## Author Comment (AC1)

Dear Referee #1,

First of all, thank you very much for the helpful comments and your kind feedback. We are very happy to read that you find it an excellent paper. Your comments are listed below, along with our responses to each one.

- Abstract "Our results show that this approach is generally better than the best randomly selected wells". This sentence must be changed because it seems to insinuate than one could change n monitoring wells randomly and they could performed better than the optimized approach (because of the world "generally').

RESPONSE: You are right, thanks for noticing this subtle detail. We will change the sentence to "Our results show that the proposed approach performs better than the best randomly selected wells"

- Introduction: "there is a dualism between monitoring costs and monitoring quality (i.e., the information gained by monitoring)" This sentence is not logical. Perhaps you mean that the higher the cost of monitoring, the better the quality of the data, provided that the monitoring is well designed. In other words, one could spend a lot of money and yet do not improve the quality of information unless the money is well spent. A deeper issue not addressed in the paper is how the expenditure in monitoring may improve the quality of groundwater, which means that groundwater monitoring is tied up to groundwater management.

RESPONSE: You are right. What we meant is that there are usually economic interests behind groundwater management and thus behind a monitoring network. As a result, many monitoring networks meet minimum requirements for groundwater management but are not scientifically sufficient to monitor the dynamics of the aquifer. We will explain this in more detail in the revised version of the paper.

- Introduction: "How does a reconstruction/interpolation error develop when a given number of monitoring wells are reduced? How does the error of reducing wells according to information content compare to a random reduction?" These sentences are confusing. Perhaps you mean: "How does the reconstruction/interpolation error varies with changes in the number of monitoring wells? How does a random reduction of monitoring wells affect the information content gained by groundwater monitoring?" I urge the authors to improve the logical meaning of their paper's text.

RESPONSE: Yes, we may have expressed ourselves somewhat misleadingly here. Thanks for your suggestion, which we will adopt only slightly modified. Our suggestion for the change would be: "How does the reconstruction/interpolation error vary as wells are progressively removed from the monitoring network in the order of the proposed ranking, and how does this compare to the removal of randomly selected wells?

- Section 2.1.4: you propose that an mxn matrix can be decomposed into matrices Q and R, such that A = QR; then you propose that there is matrix C such that A CT = Q R; unclear why not: A CT not equal to QR CT

RESPONSE: You are right. The sentence is misleading. What is meant is:

The reduced matrix QR factorization with column pivoting decomposes a matrix $\mathbf{A} \in R^{m \times n}$ into a unitary matrix $\mathbf{Q}$, an upper triangular matrix $\mathbf{R}$ and a column permutation matrix $\mathbf{C}$ **(eq. 6)** such that $\mathbf{AC^T = QR}$. We will modify the sentence accordingly

- Section 2.4.2 "Outlier values that exceeded a moving average (window size 11) of ±3 σ were removed during preprocessing" Perhaps you mean: "Data values that deviate by more than ±3 σ from the moving average (with a window size of 11 values) are considered outliers and were removed from further processing"

RESPONSE: We agree. The sentence you suggested is more comprehensible. We will gladly change it accordingly in the revised version of the paper.

- Section 2.4.3 " omnidirectional Gaussian semivariogram model" I believe you mean "an isotropic Gaussian semivariogram model"

RESPONSE: In the literature, we find both terms used as synonyms. Therefore, we think that the "omnidirectional Gaussian semivariogram model" is also correct. Maybe we could write "omnidirectional Gaussian semivariogram model (also called isotropic Gaussian semivariogram model)" to account for both terms?

- Figure 2b: the Pareto front of number of wells vs RMSE: what about Pareto fronts for the other goodness-of-fit criteria? such the NSE or the KGE?

RESPONSE: Figure 2 shows the results of GridSearchCV. Here, all combinations of the basis types, *n* basis-modes, and *n* monitoring well are tried using the k-fold-cross-validation method. The goal was to determine the most appropriate parameter combination for our task. We have chosen RMSE as the metric here, as this is the criterion used for minimization in the model. It is correct, that other goodness-of-fit criteria such as the NSE or the KGE could also be shown, but we think that would be kind of an overkill. Moreover, as can be seen in Figure 5, where the error propagation over all n monitoring wells is shown for RMSE, NSE, KGE, MAE, and $R^2$, the goodness-of-fit parameter correlate well and there are only subtle differences. The difference to figure 2b is that here the number of base modes (1043) and the base (identity) are fixed.

- Figure 7: add GMW (groundwater monitoring well) to the list of acronyms

RESPONSE: Thank you. We will add it to the acronyms.

---

## Author Comment (AC2)

Dear Referee 2

Thank you for your kind and encouraging comments on our study. Your comments and our responses to them are listed below.

- The terms "sensor" and "well" are used interchangeably throughout the manuscript. For the sake of clarity, I'd suggest sticking with only one term. The sensor could be viewed as a part of the monitoring well, therefore in my opinion it makes more sense to use "well". The goal is to optimally select wells.

RESPONSE: We agree that the frequent synonyms change disrupts the reading flow. We would propose the following solution:

Since this approach originates from standard sensor optimization and therefore, the term "sensor" is primarily used in the literature, we would retain it in the methodology section to make the comparison with other studies more straightforward. We would then use the term "well" exclusively in the results. At the beginning of the results section, we would include a sentence stating that we are applying sensor optimization to optimize groundwater monitoring wells. Therefore, the term well is used as a synonym for a sensor in the following.

- 7 and fig.6 on p.17: The performance metric nRMSE is used in Figure 6. Is it the RMSE relative to the standard deviation or the range of observations? Please explain this detail on page 7, where the RMSE equation is given.

RESPONSE: Thanks for the hint. The rRMSE used is relative to the **range** of observations. We will include the corresponding formula of rRMSE in the manuscript eq. (10).

- Lines 248-253: Although it is clear, why a single set of kriging parameters is used in the production of the map series, on what basis this particular parameter set is selected. Perhaps the values for the parameters can be provided in one additional sentence

RESPONSE: You're right. We will add the used kriging parameters (Nugget, Sill, Range) in lines 249-251 to the manuscript and the target parameter of the optimization (minimization of the mean square error) as follows: "The associated partial sill (42.70m), range (17,85m), (lag size 1,48m), nugget (0.05m) were optimized using automated CV-diagnostics to achieve the lowest mean square error.

- Line 346: It seems that reduction stages (or steps as it is used in the caption of Fig.5) refer actually to the fraction of the wells used in the analysis. My understanding from a 10% reduction is that 90% of the wells, that is 432 wells, are used. Please consider rewording or clarifying this issue where necessary.

RESPONSE: We have indeed expressed ourselves misleadingly. When we speak of a 10% reduction step in this section, the percentage refers to the remaining number of wells, so 10% means that 10% are remaining, and 90% are removed. From chapter 3.3 on, it is used in a reverse sense. We will adjust the wording in this section and caption of Fig. 5 and 6 to use "remaining subset" instead of "reduction stages." We hope this makes it more precise.

- Line 346-347: "Consequently, well 154-304-1, the highest-ranked well shown with rank 59 (bottom),…" – This is confusing because one would expect the well with rank number 1 the "most important" well.

RESPONSE: We agree and can see your point. We would change the order in Figure 6 from "most important" (top) to "least important" (bottom) as suggested.

- I'd suggest adding a little discussion in the conclusions section about the relative value of 1-D (hydrograph) data versus 2-D data. Which should be preferred if both are available? For which type of monitoring data does the presented approach work better?

RESPONSE: That is an excellent suggestion. Thank you very much for that as well! We would include your recommendation in the discussion as follows:

"Using hydrographs (1D) as input data set, the applied approach allows an information-based assessment of the operated monitoring network, with the goal of the best reconstruction of selected remaining wells. The outcomes can thus be used to identify key wells for selecting representative subnetworks, equip important wells with improved data loggers, or release installed sensors/loggers at redundant wells for more suitable locations. With two-dimensional input data, the spatial dependency structures and the area of influence of the wells can be considered  for optimizing. The goal of a 2D based reduction is thus to identify wells that are most suitable to reconstruct a spatially continuous groundwater surface. Moreover, the 2D data also allow a network extension, that is not feasible based on 1D data. Especially in combination with numerical groundwater models, we see a great synergy potential for an extension of the monitoring network tailored to the dynamics of the aquifer. When using 2d data, however, one must be aware that that the optimization approach makes suggestions based on regionalized data (either by interpolation or other methods). If these do not represent the reality sufficiently (i.e. poor regionalization), this will be also reflected in the optimization results."

- The website link for the reference on lines 559-560 needs to be corrected as it does not seem to be active

RESPONSE: We have just checked the link. It worked here. However, it takes a long time until the page is built because the file is opened directly as a pdf. We'll keep an eye on the link and change it if there are problems in the future.

---

## Author Response (AR1)

Dear Dr. Giudici,

Please find enclosed our revised manuscript. The comments/improvements of the two reviewers have been incorporated as announced in our point-by-point response of May 16, 2022. If you have any questions, please do not hesitate to contact me.

Best regards

Marc Ohmer (on behalf of the authors)